# Crystal Structure and Electrical Properties of Ruthenium-Substituted Calcium Copper Titanate

**DOI:** 10.3390/ma15238500

**Published:** 2022-11-29

**Authors:** Ljiljana Veselinović, Miodrag Mitrić, Lidija Mančić, Paula M. Jardim, Srečo Davor Škapin, Nikola Cvjetićanin, Miloš D. Milović, Smilja Marković

**Affiliations:** 1Institute of Technical Sciences of SASA, Knez Mihailova 35/IV, 11000 Belgrade, Serbia; 2The Vinča Institute of Nuclear Sciences, University of Belgrade, 11000 Belgrade, Serbia; 3Department of Metallurgical and Materials Engineering, Federal University of Rio de Janeiro, Rio de Janeiro 21941-630, Brazil; 4Jožef Stefan Institute, SI-1000 Ljubljana, Slovenia; 5Faculty of Physical Chemistry, University of Belgrade, Studentski Trg 12-16, P.O. Box 137, 11000 Belgrade, Serbia

**Keywords:** CCTO, ruthenium substituted calcium-copper titanate, dielectrics, conductors

## Abstract

This paper reports a detailed study of crystal structure and dielectric properties of ruthenium-substituted calcium-copper titanates (CaCu_3_Ti_4−*x*_Ru*_x_*O_12_, CCTRO). A series of three samples with different stoichiometry was prepared: CaCu_3_Ti_4−*x*_Ru*_x_*O_12_, *x* = 0, 1 and 4, abbreviated as CCTO, CCT3RO and CCRO, respectively. A detailed structural analysis of CCTRO samples was done by the Rietveld refinement of XRPD data. The results show that, regardless of whether Ti^4+^ or Ru^4+^ ions are placed in *B* crystallographic position in *AA*’_3_*B*_4_O_12_ (CaCu_3_Ti_4−*x*_Ru*_x_*O_12_) unit cell, the crystal structure remains cubic with $Im\bar{3}$ symmetry. Slight increases in the unit cell parameters, cell volume and interatomic distances indicate that Ru^4+^ ions with larger ionic radii (0.62 Å) than Ti^4+^ (0.605 Å) are incorporated in the CaCu_3_Ti_4−*x*_Ru*_x_*O_12_ crystal lattice. The structural investigations were confirmed using TEM, HRTEM and ADF/STEM analyses, including EDXS elemental mapping. The effect of Ru atoms share in CaCu_3_Ti_4−*x*_Ru*_x_*O_12_ samples on their electrical properties was determined by impedance and dielectric measurements. Results of dielectric measurements indicate that one atom of ruthenium per CaCu_3_Ti_4*−x*_Ru*_x_*O_12_ unit cell transforms dielectric CCTO into conductive CCT3RO while preserving cubic crystal structure. Our findings about CCTO and CCT3RO ceramics promote them as ideal tandem to overcome the problem of stress on dielectric-electrode interfaces in capacitors.

## 1. Introduction

The CaCu_3_Ti_4_O_12_ (CCTO) materials belong to a large family of the *AA*’_3_*B*_4_O_12_ *A*-site ordered perovskites. During the last two decades, CCTO materials have been extensively investigated due to very high dielectric permittivity (up to 10^5^), which is practically constant in a broad temperature (100 and 600 K) and frequency (1 kHz–1 MHz) range [1,2,3,4]. Due to such characteristics, CCTO ceramics attract the attention of researchers as very promising materials for applications in microelectronics and microwave devices [5]. The giant dielectric permittivity of CCTO materials promotes their utilization for the fabrication of capacitors with improved capacitive performance for portable electronic devices [6]. Though, it was observed that the nature of the ceramic-electrode interface could affect the decrease of the dielectric permittivity. Significant differences in the crystal structure and electrical properties of the CaCu_3_Ti_4_O_12_ as a dielectric material and a metallic electrode cause an energy barrier and the occurrence of stress on the ceramic-electrode interface, which reduces the dielectric permittivity. Therefore, for the fabrication of capacitors with good capacitive performance, it is important to combine dielectric ceramic and electrodes with similar crystal structures and unit cell parameters [7]. The reduction of the stress on the ceramic-electrode interfaces can be achieved by using commercially available materials as an interlayer having a close lattice parameter match with both dielectric and electrode. It was noticed that the incorporation of Ru^4+^ ions in the CCTO crystal structure dramatically increases the conductivity of these materials [8,9,10,11]. The CaCu_3_Ru_4_O_12_ (CCRO) materials are isostructural with CaCu_3_Ti_4_O_12_ materials with cubic $Im\bar{3}$ space group showing the Pauli-paramagnetic and metallic character [12]. With the CCRO material as an interface between CCTO ceramic and metallic electrodes, it is possible to reduce the stress on the dielectric-electrode interfaces.

A cubic perovskite-structure oxide with the general chemical formula *AA*’_3_*B*_4_O_12_ is derived from that of a simple perovskite, *AB*O_3_ type, structure. The crystal structure of *AB*O_3_ perovskite is very flexible, enabling a wide range of ionic substitutions into both *A* and *B* crystallographic positions giving the possibility of creating new compounds with new functional properties. However, the *AA*’_3_*B*_4_O_12_ structural type is more rigid than the parent *AB*O_3_. This structure was induced by a+a+a+ (Glazer’s nomenclature), an octahedral tilting distortion of BO_6_ octahedra resulting in the B–O–B angle varying considerably away from 180^o^, which is typical for the ideal cubic perovskite structure [13,14,15,16]. Such octahedral distortion produces a structure where three-fourths of the *A*-site cations have square planar coordination (*A*’-cation site), while the remaining one-fourth of the *A*-site cations retain 12-fold coordination. The lattice parameters are twice as long as the ideal perovskite cubic unit cell (V = 2ap·2ap·2ap). The *A*’- cation site is occupied by Cu^2+^ or Mn^3+^ ions, while the crystallographic *A* position is usually occupied by larger ions, such as alkaline-earth ions or lanthanides. The *B* cation determines the electronic and magnetic properties of the materials [17,18]. Titanium ions in the *B* position (in the CaCu_3_Ti_4_O_12_ compounds) contribute to the high dielectric permittivity and better semiconductor properties. Over the years, different preparation techniques have been used to improve the dielectric properties of CCTO ceramics as well as metal ion substitution, both single and co-doped. It has been reported that metal ions such as Mg [19], Sn [20], Ge [21], Zr [22], and a combination of Zn and Mn [23], Y and Zr [24], or Mg with Ge [25] can noticeably reduce the dielectric loss (tgδ) and lead to the retention of a very high value for the dielectric constant (ε’).

On the contrary, our interest was to synthesize CaCu_3_Ti_4−*x*_Ru*_x_*O_12_ as a compound isostructural with CaCu_3_Ti_4_O_12_ but with much larger conductivity which can be successfully used as an interlayer between dielectric ceramic and metallic electrode with the aim to reduce interlayer stress. Even though Ru^4+^ ions in the *B* position dramatically increase the conductivity of *AA*’_3_*B*_4_O_12_ materials, ruthenium is not acceptable for commercial application due to its highcost.

The aim of our work was to optimize the amount of ruthenium in the CaCu_3_Ti_4−*x*_Ru*_x_*O_12_ crystal structure in order to obtain inexpensive and commercially acceptable materials with the desired functional properties. A series of CaCu_3_Ti_4−*x*_Ru*_x_*O_12_ (*x* = 0, 1, and 4) ceramics was synthesized by the semi-wet precipitation method. A detailed structural analysis was done by the Rietveld refinement of X-ray powder diffraction data. In order to confirm the results of the structural investigations, TEM, HRTEM, and ADF/STEM analysis with EDXS elemental mapping were performed. To comprehend the influence of ruthenium amount in the CaCu_3_Ti_4−*x*_Ru*_x_*O_12_ crystal structure on its electrical properties, we employed impedance spectroscopy and dielectric measurements.

## 2. Materials and Methods

Semi-wet precipitation method was used for the synthesis of CCTRO with three different stoichiometries: CaCu_3_Ti_4−*x*_Ru*_x_*O_12_; *x* = 0, 1 and 4, abbreviated as CCTO, CCT3RO, and CCRO, respectively. Semi-wet precipitation method was chosen for the preparation of the materials since, compared with the solid-state technique, it enables better homogenization of the starting reagents. In a typical synthesis, a weighed amount of Ca(CH_3_COO)_2_ × H_2_O (99%, Alfa Aesar) and Cu(NO_3_)_2_ × 2.5H_2_O (99.99%, Aldrich, St. Louis, MO, USA) were dissolved in 20 mL of distilled water. A source of Ti^4+^ was Ti[OCH(CH_3_)_2_]_4_ (97%, Aldrich Chemical Company Inc.), which was dissolved in 10 mL of acidified water. The stable solutions were mixed, and, in this solution, a RuO_2_ (99.9%, Aldrich) was dispersed. The prepared dispersion was evaporated in a silicon oil bath while mixing with a magnetic stirrer. Obtained dry powder was homogenized with an agate mortar and pestle and consecutively calcined at 700 °C. The resulting powder was milled in planetary YTZ ball-mil for 0.5 h in ethanol and fired in the air at 900 °C for 20 h.

The structural investigations were performed by the Rietveld refinement [19] of the XRD data recorded at room temperature on a Philips PW 1050 diffractometer with CuKα1,2 (λ = 1.54178 Å) Ni-filtrated radiation. The diffraction intensity was measured from 10 to 130° 2θ, using a step size of 0.02° with a counting time of 12 s per step. The working conditions were 40 kV and 20 mA. The Rietveld refinement of the CaCu_3_Ti_4−*x*_Ru*_x_*O_12_ crystal structure was performed using the FullProf computing program in the WinPLOTR environment [26,27,28,29]. The unit cell parameters were calculated using the program LSUCRI (least squares unit cell refinement with indexing) [30]. The Rietveld refinement was started from the calculated unit cell parameters and previously reported atomic positions [15]. A TCH-pseudo-Voigt function was chosen for the line shape of the diffraction peaks, whereas a 6-coefficient polynomial function was used for the background description. Transmission electron microscopy (TEM) and ADF/STEM with EDXS elemental mapping were performed using Jeol JEM 210 TEM and Titan G2 80-200 with ChemiSTEM technology. The fast Fourier transformation (FFT) analysis was done in the Digital Micrograph computer program.

The synthesized CCTRO powders were uniaxially pressed in a die (∅ 10 mm) under the pressure of about 400 MPa. Each compact has a thickness of *circa* 3 mm and 60 ± 2% of theoretical density. Sintering of the green compacts was carried out in a Protherm tube furnace in an air atmosphere, with a heating rate of 5°/min up to 1050 °C and with a dwell time of 12 h. The electrical measurements were done on the sintered CCTRO pellets. The capacitance and conductivity measurements were performed in function of temperature at a frequency of 1 kHz (internal frequency of the instrument) at the Wayne Kerr Universal Bridge B224. All measurements were done in the cooling mode from 150 to 21.3 °C in an air atmosphere. In order to improve the contact between the silver electrodes and the surface of the samples, both sample bases are coated with a suspension of silver powder and ethyl acetate. Impedance measurements were performed on the Potentiostat-Galvanostat PAR 273A coupled with Dual Phaslock-in Amplifier 5210 brand. The measurements were done in the frequency interval from 0.01 Hz to 100 kHz, and the temperature range from 150 °C to room temperature. Mathematical modeling of experimentally obtained data was done by Z-View2 (version 2.6 demo) computer program.

## 3. Results and Discussion

The structural refinement of room temperature XRD data of three investigated powders: CaCu_3_Ti_4_O_12_ (CCTO), CaCu_3_Ti_3_RuO_12_ (CCT3RO) and CaCu_3_Ru_4_O_12_ (CCRO), were performed in the space group $Im\bar{3}$. The Rietveld analysis was done by using the FullProf computing program. A pseudo-Voigt function was chosen to generate the line shape of the diffraction maximums. The scale factor, peak-shape parameters, lattice parameters, variable fractional atomic coordinates, and displacement parameters were refined. The unit cell parameters of the CCTO materials are doubled in all three crystallographic axes (V = 2ap·2ap·2ap) compared to the ideal (*AB*O_3_) cubic perovskite lattice (ap). The Rietveld refinements were started from the previously reported atomic positions: Ca atoms were situated in 2a Wyckoff sites at the (0 0 0) position, Cu atoms occupied the 6b at (0 ½ ½), Ti in 8c (0 ¼ ¼), and O in 24g (x y 0) sites. The literature data [15] of the positional coordinates were used as starting values for structural refinement. Figure 1 illustrates the good agreement between calculated and observed structural models. The obtained results confirm that the powders are a single-phase perovskite oxide without a secondary phase.

The structural refinement was started by allowing varying the values of the zero point, the scale factor, and the background coefficients. Then, lattice parameters, asymmetry, and half-width parameters (U, V, W) were refined. When satisfactory values of profile parameters were obtained, the structural parameters, i.e., atomic positional parameters and isotropic displacement parameters, were allowed to vary. The overall displacement parameter was refined for every atom; the obtained value was set as starting isotropic displacement parameter for each of the atoms. The isotropic displacement parameters for Ca, Cu, Ti and Ru atoms were refined independently, while for the O atoms, the displacement parameters were constrained at the same value. The titanium and ruthenium atoms in the CaCu_3_Ti_4−*x*_Ru*_x_*O_12_ crystal structure are situated in the same crystallographic positions, so their fractional atomic coordinates were set to be equal during refinement. Their occupational factors are interrelated, and only the Ti/Ru ratio was allowed to vary, keeping the overall occupancy of the respective sites constant according to the appropriate stoichiometric ratio. The refined values of the unit cell parameters, atomic position parameters, and displacement parameters are listed in Table 1.

The refined values of the unit cell parameters and unit cell volume increase with increases in ruthenium ions content in the crystal structure of the investigated powders (Table 1, Figure 2). The increases in unit cell parameters are provoked by partial (1 of 4) or complete (4 of 4) replacement of Ti^4+^ ions (0.605 Å ionic radii for six coordination) by larger Ru^4+^ ions (0.62 Å ionic radii for six coordination) in the CaCu_3_Ti_4−_*_x_*Ru*_x_*O_12_ crystal structures.

The interatomic distances in the CaCu_3_Ti_4_O_12_, CaCu_3_Ti_3_RuO_12_ and CaCu_3_Ru_4_O_12_ powders were calculated from the structural results obtained by Rietveld refinement and listed in Table 2. The increase of the average value of the interatomic Ti/Ru-O distance from 1.953 Å (CCTO) to 1.967 Å (CCRO) indicates the substitution of smaller Ti^4+^ by larger Ru^4+^ cation in the CaCu_3_Ti_4−_*_x_*Ru*_x_*O_12_ crystal structure.

The results of the structural investigation were confirmed by TEM and SAED analyses. TEM and SAED analyses of CCTO and CCRO powders are presented in Figure 3. Figure 3a reveals significant agglomeration of CCTO single-crystalline particles whose size is about several tens of nanometers (inset in Figure 3a). SAED pattern, presented in Figure 3b, confirms crystallographic information obtained through Rietveld refinement. Interplanar distances of 3.701, 2.597, 1.837, 1.516, and 1.253 Å correspond well too (200), (220), (400), (422), and (433) planes of CaCu_3_Ti_4_O_12_ phase, JCPDS 75-2188. Similarly, many randomly oriented CCRO particles are presented in Figure 3c. Their single-crystalline nature is exposed in Figure 3d, which shows the overlapping of crystals with 2 different zone axes with low mismatching. Observed interplanar distances of 3.740, 1.860, 2.630, and 1.320 Å are assigned to (200), (400), (220), and (440) planes of CaCu_3_Ru_4_O_12_ phase, JCPDS 41-0821.

The TEM, HRTEM, and FFT analysis of the CCT3RO sample presented in Figure 4 confirm the cubic crystal symmetry with $\mathrm{the}\;Im\bar{3}$ space group. Crystallite viewed in the [−111] zone axis as well as two sets of crystallographic planes (400) and (220) with interplanar distances of 1.8 and 2.6 Å, respectively, agree with a literature data for cubic CaCu_3_Ti_4_O_12_ (JCPDS 75-2188).

High-angle annular dark-field STEM and corresponding EDXS elemental mapping presented in Figure 5 implies the homogeneous distribution of Ca, Cu, Ti, Ru and O in CCT3RO particles, independently of their size.

Complex impedance spectroscopy (IS) was used to determine the electrical properties of sintered CCTRO ceramics. Recorded Nyquist plots are presented in Figure 6.

The recorded complex impedance spectra of CCTRO sintered ceramics were fitted using the Z-View2 computing program according to the equivalent circuit. The equivalent circuit consisting of one resistor (R) and one capacitor (constant phase element, CPE) connected in parallel best describes the impedance spectra of CCTRO materials. The calculated resistance values are in the range of 1 to 150 MΩ. Relatively high values of resistance indicate good dielectric properties of CCTO material. On the contrary, the impedance spectra of sintered ceramics with ruthenium content show an accumulation of points for higher and middle-frequency values of about 0.1 Ω and 0.05 Ω for CCT3R and CCRO, respectively. Only at very low-frequency values, from 0.1 to 0.01 Hz, the impedance spectra of these ceramics can be described by an equivalent circuit of one resistor (R) and one capacitor (CPE) connected in parallel. However, the difference in the obtained resistivity values for CCTRO materials with and without ruthenium content is more than eight orders of magnitude in the *direct current circuit*. Such behavior indicates good conductivity of CCT3RO and CCRO ceramics. The electrical conductivity of investigated ceramics at four characteristic frequency points is presented in Table 3.

The difference in electrical conductivity can be explained by different electron configurations of Ti^4+^ and Ru^4+^ ions [14,31,32]. Titanium is a 3*d* element with an electronic configuration:Ti: 1s^2^ 2s^2^ 2p^6^ 3s^2^ 3p^6^ 4s^2^ 3d^2^

However, the electronic configuration of Ti^4+^ cation in the CCTO crystal structure turns into:Ti^4+^: 1s^2^ 2s^2^ 2p^6^ 3s^2^ 3p^6^
with closed external 3p^6^ shells, which explains the relatively low conductivity of these materials. The isomorphic replacement of the Ti^4+^ ions by Ru^4+^ leads to different electronic configurations. Explicitly, Ru^4+^ has the following electronic configuration:Ru^4+^: 1s^2^ 2s^2^ 2p^6^ 3s^2^ 3p^6^ 4s^2^ 3d^10^ 4p^6^ 5s^0^ 4d^4^,
with uncompleted *d* subshells. These four electrons in the octahedral crystalline environment are situated in *t_g_* crystalline orbitals containing two unpaired *d* electrons, i.e., Ru^4 +^ represents a magnetic ion in the CCRO crystal. The CCRO conductivity can be explained by the Kondo mechanism. Namely, CCRO materials are *d*-electron heavy-fermion system, which behaves similarly to an *f*-electron heavy-fermion system. In CCRO structures, the Cu 3*d* electrons are localized, while 4*d* electrons of magnetic Ru^4+^ ions contribute to the conductivity [14,32,33].

Figure 7 displays the complex impedance spectra (Nyquist plot) of CCTO ceramic measured in an air atmosphere, in the temperature range from 150 to 23 °C (room temperature) and frequency interval from 100 kHz to 0.01 Hz. The obtained impedance diagrams represent the sum of the grain-interior and grain boundary resistance, which is impossible to separate. Thus, the fitting of summary data was done. The fitting was achieved based on the corresponding equivalent circuit (inset in Figure 7) and the *Z-View*2 computing program.

The values of electric resistivity determined by fitting with an appropriate equivalent circuit were used to calculate the values of electrical resistivity ρ [Ω⋅cm] and specific electrical conductivity σ (S·cm^−1^). The calculated values are presented in Table 4. The calculated values of electrical conductivity (Table 4) were used to determine the activation energies *E*_a_ based on the Arrhenius equation, Equation (1) [34,35]:(1)$$\sigma =\sigma_{o}exp\left(-\frac{E_{a}}{k_{B}T}\right)$$
where *E*_a_ is the activation energy, *k*_B_ is the Boltzmann’s constant (8.8617343 × 10^−5^ eV/K), and *σ*_o_ is a constant. The obtained relationship is almost linear (Figure 8). The calculated value of activation energy of 0.49 eV is in accordance with literature data [1,2,3].

Figure 9 represents the influence of both the temperature and the frequency on the dielectric properties of the CCTO ceramic. The measurements were done in the frequency interval from 100 kHz to 0.01 Hz and in the temperature range between 177 and 23 °C.

It is obvious that CCTO materials possess a high dielectric permittivity *ε*’ (>10^3^) which significantly depends on temperature and frequency (Table 5).

The influence of temperature and frequency on the dielectric loss (tg*δ*) of the CCTO ceramic is shown in Figure 10. For the three highest temperatures, the dielectric loss monotonically decreases with increasing frequency. For the three lowest temperatures, the dielectric losses slightly increase in the frequency range from 10 to 100 kHz.

The relative dielectric constant *ε_r_* for CCTO material was measured in cooling mode from 150 °C to room temperature at a constant frequency of 1 kHz. The relative dielectric constant increases with increasing temperature from about 1460 at room temperature to 2460 at 150 °C (Figure 11a). It can be seen that these measurements are in good agreement with the values obtained from the impedance measurements at the same frequency of 1 kHz (Table 5).

The dependence of dielectric loss (tg*δ*) on temperature is represented in Figure 11b. It can be seen that the tg*δ* increases with an increase in the temperature. These values are in the range between 0.13 at room temperature and 0.65 at 150 °C, which is in good agreement with the values obtained at 1 kHz (Figure 10).

The results of electrical measurements clearly show that the stoichiometry of CaCu_3_Ti_4−_*_x_*Ru*_x_*O_12_ significantly affects their electrical properties. It is known that CaCu_3_Ti_4_O_12_ material exhibits excellent dielectric properties with a dielectric constant in excess of 10^4^. Quite the contrary, the CaCu_3_Ru_4_O_12_ material exhibits conductive behavior [7,36,37,38]. The results presented in this paper revealed that the substitution of one Ti^4+^ ion by Ru^4+^ per the CaCu_3_Ti_4_O_12_ unit cell changes the electrical characteristics of these materials from dielectric to conductive. Such electrical properties of these materials are very important for their potential application in the electronics industry for capacitors production. However, in these components, the problem of stress on the dielectric-electrode interfaces occurs due to the different structural characteristics of the metal (electrode) and ceramic materials. This problem can be overcome by the use of materials that would have the same or similar structural characteristics both with dielectric and with an electrode, which would be used as an interface. Potential materials that could be used for the production of a contact or interlayer are CCT3RO and CCRO, owing to the same crystal structure and close lattice parameters (Table 1).

## 4. Conclusions

In this paper, we presented a detailed analysis of crystal structure and dielectric properties of three CaCu_3_Ti_4−*x*_Ru*_x_*O_12_ (*x* = 0, 1 and 4) samples. The detailed analysis of the crystal structure was determined by Rietveld’s refinement of XRD data. The results of the structural analysis show that regardless of whether Ti^4+^ or Ru^4+^ ions are placed in *B* crystallographic position in the CaCu_3_Ti_4−*x*_Ru*_x_*O_12_ unit cell, the crystal structure remains cubic with $\mathrm{the}\;Im\bar{3}$ space group. Slight increases in the unit cell parameters, cell volume and interatomic distances indicate that Ru^4+^ ions with larger ionic radii (0.62 Å) than Ti^4+^ (0.605 Å) are incorporated in the CaCu_3_Ti_4−_*_x_*Ru*_x_*O_12_ crystal lattice. Results of TEM, HRTEM and ADF/STEM analyses with EDXS elemental mapping confirmed the results of structural investigations. The electrical properties of the CaCu_3_Ti_4−*x*_Ru*_x_*O_12_ materials were investigated based on impedance spectroscopy and dielectric measurements. Values of electrical resistance measured at room temperature indicate that the ruthenium ions in the CaCu_3_Ti_4_O_12_ structure change the electrical properties of these materials from dielectric to conductive. What is more, just one atom of ruthenium per unit cell is enough to drastically improve the conductive properties of these materials. With the addition of ruthenium ions, conductivity increases for more than five orders of magnitude for the frequency of 100 kHz and for almost eight orders of magnitude for the frequency of 100 Hz.

Our findings about CCTO and CCT3RO ceramics, as both an inexpensive and ideal tandem to overcome the problem of stress on dielectric-electrode interfaces in capacitors, promote their commercial application in the electronic industry.

## Figures and Tables

**Figure 1 materials-15-08500-f001:**
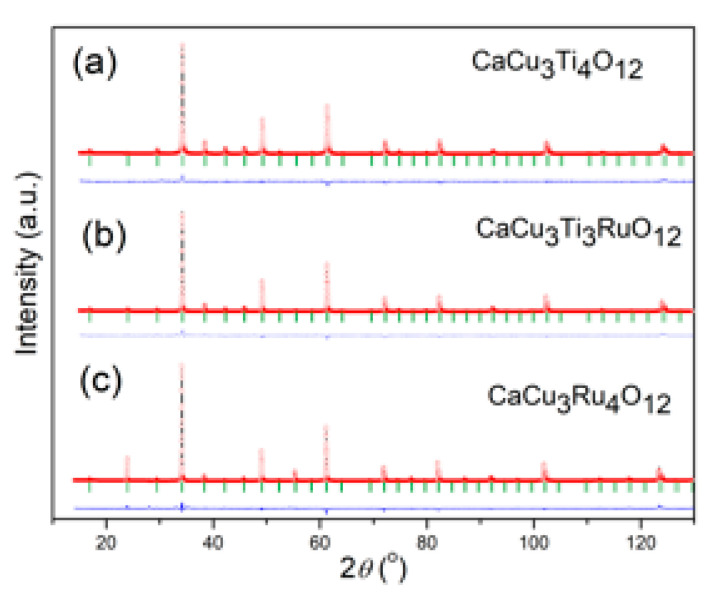
The Rietveld refinement plot of (**a**) CaCu_3_Ti_4_O_12_ (CCTO), (**b**) CaCu_3_Ti_3_RuO_12_ (CCT3RO), and (**c**) CaCu_3_Ru_4_O_12_ (CCRO) samples.

**Figure 2 materials-15-08500-f002:**
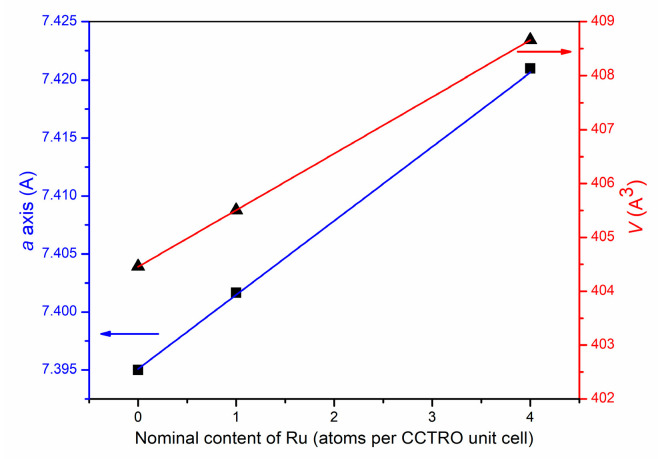
Relationship between the unit cell parameters and cell volume with a nominal content of Ru atoms per CCTRO unit cell.

**Figure 3 materials-15-08500-f003:**
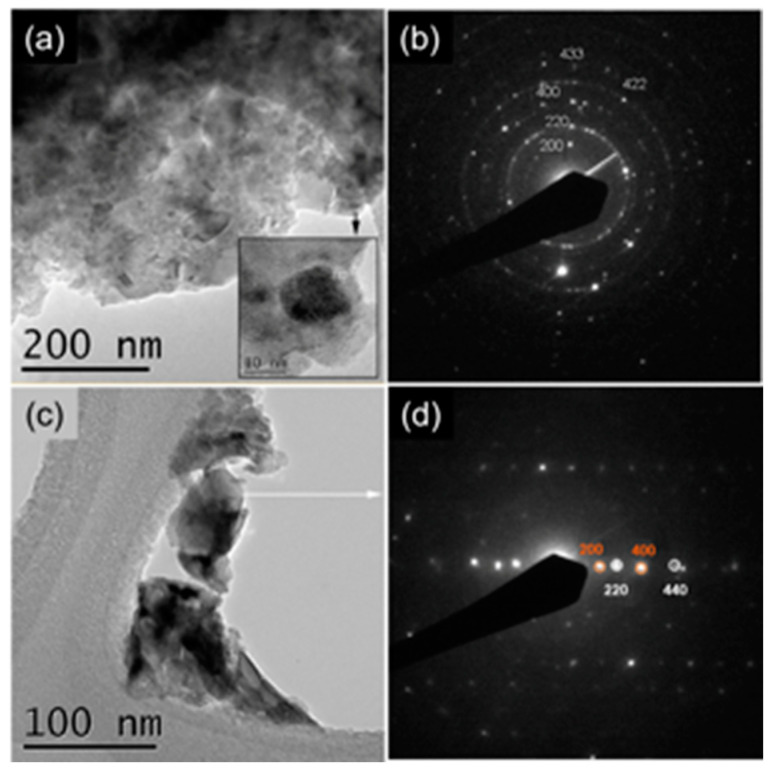
TEM and SAED of (**a**,**b**) CCTO and (**c**,**d**) CCRO samples.

**Figure 4 materials-15-08500-f004:**
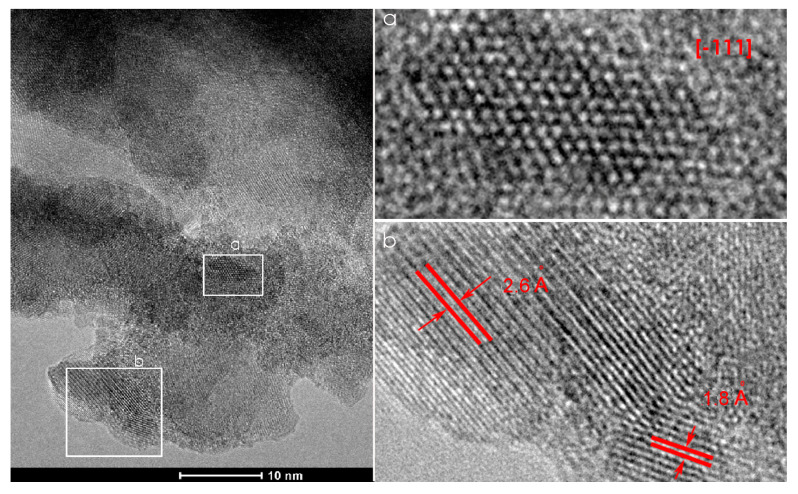
TEM (left) and HRTEM (**a**,**b**—right) of CCT3RO sample.

**Figure 5 materials-15-08500-f005:**
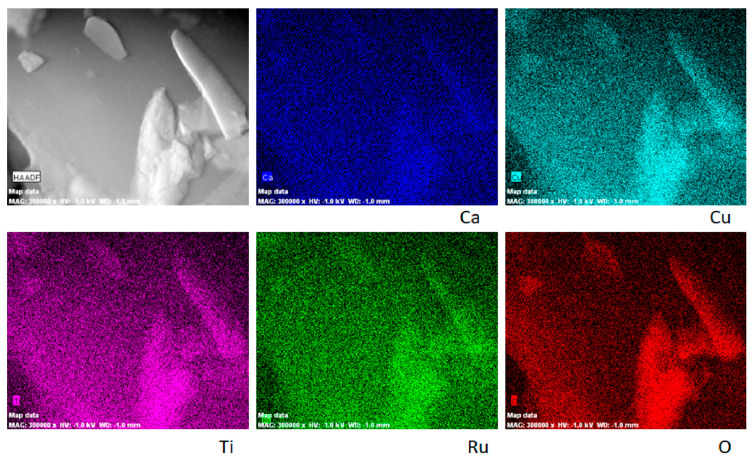
ADF/STEM analysis and EDXS elemental mapping of CCT3RO powder.

**Figure 6 materials-15-08500-f006:**
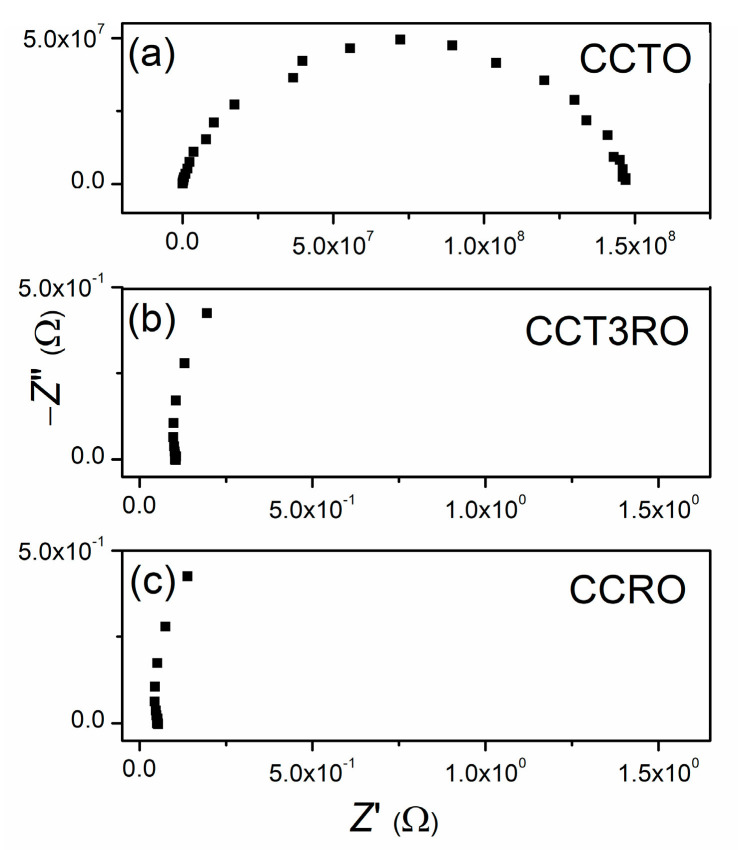
Complex impedance spectra recorded at room temperature for sintered ceramics: (**a**) CCTO, (**b**) CCT3RO, and (**c**) CCRO.

**Figure 7 materials-15-08500-f007:**
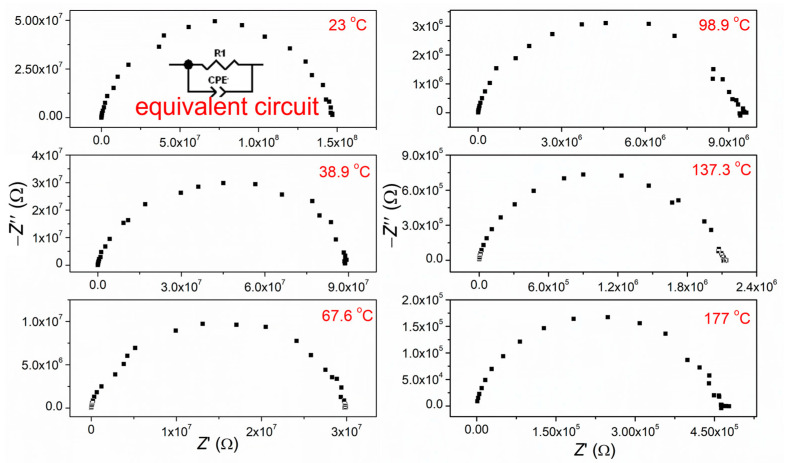
Complex impedance spectra for CCTO ceramics at selected temperature; as an inset: scheme of an equivalent circuit model used for fitting impedance spectra of CCTO ceramic.

**Figure 8 materials-15-08500-f008:**
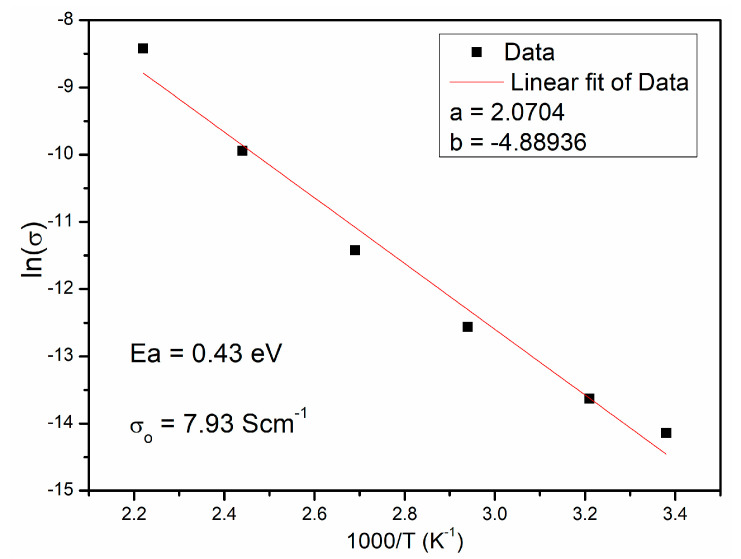
The Arrhenius plot of conductivity ln(σ) versus 1000/T of CCTO ceramic.

**Figure 9 materials-15-08500-f009:**
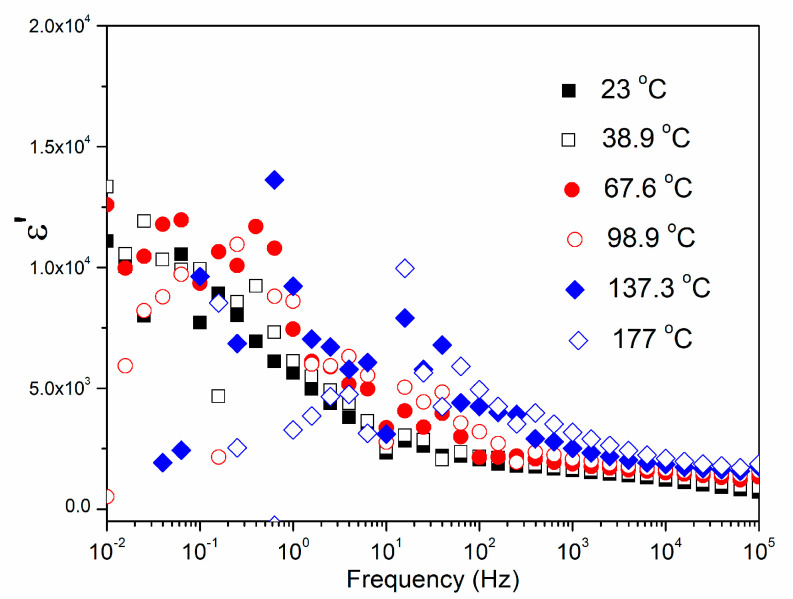
Frequency−dependent dielectric constant *ε*’ of the CCTO ceramic at different temperatures.

**Figure 10 materials-15-08500-f010:**
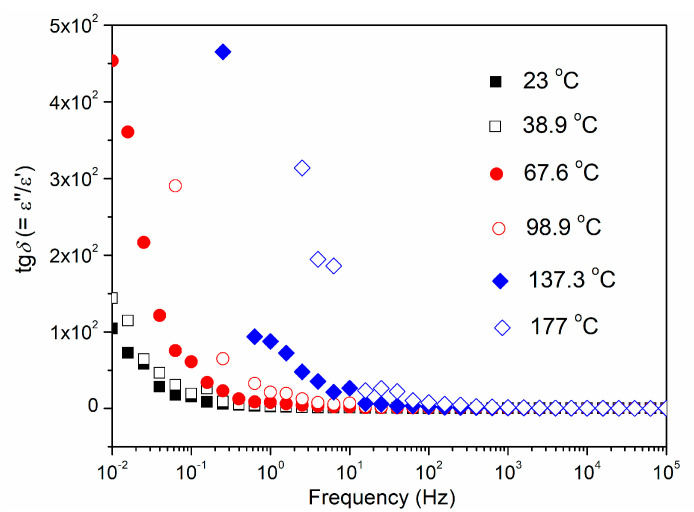
Frequency−dependent dielectric loss (tg*δ*) of the CCTO ceramic at different temperatures.

**Figure 11 materials-15-08500-f011:**
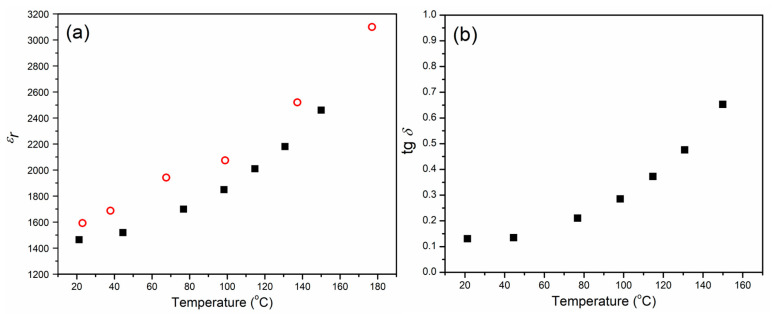
The temperature dependence of (**a**) relative dielectric constant was calculated based on impedance measurements (circle) and based on measured capacitance (square), and (**b**) dielectric loss for CCTO ceramic.

**Table 1 materials-15-08500-t001:** The refined values of the fractional coordinates and isotropic displacement parameters of the CaCu_3_Ti_4−*x*_Ru*_x_*O_12_ powders.

Refined Parameters	Investigated Powders Space Group		
	CaCu_3_Ti_4_O_12_ $Im\bar{3}$	CaCu_3_Ti_3_RuO_12_ $Im\bar{3}$	CaCu_3_Ru_4_O_12_ $Im\bar{3}$
Lattice parameters			
*a* (Å)	7.3954(2)	7.4016(5)	7.4209(6)
*V* (Å^3^)	404.46(8)	405.50(5)	408.66(5)
Refined fractional coordinates and isotropic displacement parameters *B* (Å^2^)			
*x*	0.0	0.0	0.0
*y*	0.0	0.0	0.0
*z*	0.0	0.0	0.0
*B*	1.082(8)	1.36(3)	0.76(9)
Cu			
*x*	0.5	0.5	0.5
*y*	0.5	0.5	0.5
*z*	0.0	0.0	0.0
*B*	1.974(3)	0.974(3)	0.76(6)
Ti/Ru			
*x*	0.25	0.25	0.25
*y*	0.25	0.25	0.25
*z*	0.25	0.25	0.25
*B*	0.657(5)	0.657(5)	0.76
O			
*x*	0.3043(8)	0.3044(4)	0.3063(7)
*y*	0.1805(8)	0.1817(5)	0.1839(7)
*z*	0.0	0.0	0.0
*B*	0.775(3)	0.942(5)	0.76(3)

**Table 2 materials-15-08500-t002:** The interatomic distances (Å) and reliability factors of CaCu_3_Ti_4−_*_x_*Ru*_x_*O_12_ powders.

Refined Parameters	Investigated Powders Space Group		
	CaCu_3_Ti_4_O_12_ $Im\bar{3}$	CaCu_3_Ti_3_RuO_12_ $Im\bar{3}$	CaCu_3_Ru_4_O_12_ $Im\bar{3}$
Interatomic distances (Å)			
Ca―O	2.618(5)	2.623(7)	2.649(5)
Cu―O	1.960(4)	1.985(4)	1.985(4)
Ti/Ru―O	1.953(5)	1.957(5)	1.965(5)
Agreement factors (%)			
*Rwp*	16.0	13.7	14.50
*RB*	10.9	12.6	9.75
*Χ2*	0.9	2.72	3.55

**Table 3 materials-15-08500-t003:** Dependence of the electrical conductivity and the frequency value at room temperature.

Frequency (kHz)	σ (S·cm^−1^)		
	CCTO	CCT3RO	CCRO
100	1.1 × 10^−4^	6.0	7.1
10	2.6 × 10^−5^	11.6	21.0
1	5.0 × 10^−6^	11.3	19.0
0.1	4.0 × 10^−7^	11.2	18.6

**Table 4 materials-15-08500-t004:** Change in the value of specific electrical resistance and specific electrical conductivity with temperature change for CCTO.

Temperature (°C)	Specific Electrical Resistivity (Ω·cm)	Specific Electrical Conductivity (S·cm^−1^)
177.0	4.85 × 10^5^	0.22 × 10^−3^
137.3	2.17 × 10^4^	0.48 × 10^−4^
98.9	9.39 × 10^4^	0.11 × 10^−4^
67.6	2.72 × 10^5^	0.35 × 10^−5^
38.9	8.96 × 10^5^	0.12 × 10^−5^
23.0	1.47 × 10^6^	0.72 × 10^−6^

**Table 5 materials-15-08500-t005:** The frequency and temperature dependence of the dielectric constant (ε’) and the dielectric loss (tgδ) for CCTO ceramic.

Frequency (kHz)	Temperature (°C)					
	23.0	38.9	67.6	98.9	137.3	177.0
	*ε*’					
100	712.7	916.4	1330.2	1569.2	1753.9	1867.9
10	1295.9	1344.3	1507.3	1698.5	1849.9	2093.9
1	1597.9	1687.0	1875.2	2868.6	2519.8	3178.8
0.1	2050.9	2170.0	2140.3	3195.5	4249.7	4950.5
	tg*δ*					
100	0.413	0.343	0.220	0.153	0.115	0.134
10	0.267	0.219	0.169	0.167	0.237	0.406
1	0.184	0.183	0.230	0.332	0.643	1.484
0.1	0.310	0.402	0.750	0.978	2.296	7.681

## Data Availability

Not applicable.

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
