# Peer review of "Crystal Structure and Electrical Properties of Ruthenium-Substituted Calcium Copper Titanate"

_materials, 2022, doi:10.3390/ma15238500_

Round 1

Reviewer 1 Report

This work reports a study of crystal structure and electric properties of ruthenium-substituted calcium-copper titanates. The authors performed an extensive investigation and shown several experimental data. The manuscript can be published after some revisions. In particular, writing style should be improved as well as grammar and syntax. In addition, several typos are present throughout the text. Please, find in what follow other specific comments.

1.     Page 3, line 112 and Page 7, line 211: Why is here mentioned only the CCTRO system?

2.   Page 5, lines 171-177: This part is redundant. Figure 2 already shows this information.

3.       Page 6, lines 187 and 192: It is not clear the meaning of “single-crystalline”.

4.     Figure 4: I suggest checking the reported interlayer distance. These values appear inverted.

5.     Table 4: Data are reported twice.

6.     Figure 8: Value of sigma0 should be also reported.

7.    Figure 8. Reported values of sigma do not match with Table 4.

8.     Figure 9: the y-axis should be rescaled, i.e., epsilon_prime up to 2x104 for instance.

9.     Page 10, lines 280-283. Statement and conclusion are not supported by the data reported in Table 5.

--   Figure 10: It is not specified in the text what material this figure is related to.

Author Response

We are thankful for positive evaluation and a recommendation from the Reviewer 1. According to the suggestions of Reviewer 1 we made all necessary corrections.

This work reports a study of crystal structure and electric properties of ruthenium-substituted calcium-copper titanates. The authors performed an extensive investigation and shown several experimental data. The manuscript can be published after some revisions. In particular, writing style should be improved as well as grammar and syntax. In addition, several typos are present throughout the text. Please, find in what follow other specific comments.

Point 1:. Page 3, line 112 and Page 7, line 211: Why is here mentioned only the CCTRO system?

Response 1: CCTRO is the abbreviation for CaCu3Ti4-xRuxO12, thus includes all examined powders, while CCTO, CCT3RO and CCRO are used for certain compounds where x = 0, 1 and 4, respectively. Beside on Page 3, line 113 and Page 7, line 211, the abbreviation CCTRO is used many times in the manuscript.

Point 2: Page 5, lines 171-177: This part is redundant. Figure 2 already shows this information.

Response 2: According to the Reviewer 1 suggestion text from the Page 5, lines 171-177 is removed from the revised manuscript.

Point 3: Page 6, lines 187 and 192: It is not clear the meaning of “single-crystalline”.

Response 3: We used term single-crystalline as a synonym for term monocrystalline. Actually, the idea was to emphasize agglomeration of small monocrystalline particles (inset in Figures 3a and 3c).

Point 4: Figure 4: I suggest checking the reported interlayer distance. These values appear inverted.

Response 4: The interlayer distances reported in Fig 4c were checked and by omission inverted values were reported. Figure 4 with corrected interlayer distances is provided in the revised manuscript.

Point 5: Table 4: Data are reported twice.

Response 5: We are thankful for this observation. In the revised manuscript Table 4 is corrected.

Point 6: Figure 8: Value of sigma0 should be also reported.

Response 6: In the revised manuscript value of sigma0 is denoted in Figure 8.

Point 7: Figure 8. Reported values of sigma do not match with Table 4.

Response 7: The authors made an omission. Figure 8 which match with the values presented in Table 4 is included in the revised manuscript.

Point 8: Figure 9: the y-axis should be rescaled, i.e., epsilon_prime up to 2x104 for instance.

Response 8: Figure 9 is revised according to the Reviewer 1 suggestion; the y-axis is rescaled to 2x104.

Point 9: Page 10, lines 280-283. Statement and conclusion are not supported by the data reported in Table 5.

Response 9: The statement from Page 10, lines 280-283 is excluded from the revised manuscript.

Point 10: Figure 10: It is not specified in the text what material this figure is related to.

Response 10: The authors are thankful to the Reviewer 1 for suggestions. The figure 10 shows the influence of temperature and frequency on the dielectric loss (tgδ) for CCTO ceramic, which is additionally stated in the text and in the figure caption.

Reviewer 2 Report

The results are of potential interest to readers in related areas. The manuscript is recommended for publication after addressing the following issues.

(a) The manuscript should be polished in language.

(b) Electronic structructures of elements/ions could be removed.

(c) Equations/formulas should be numbered.

(d) Some figures should be enlarged.

Author Response

We are thankful for a detailed analysis of our results and constructive suggestions which encourage us to improve the manuscript.

The results are of potential interest to readers in related areas. The manuscript is recommended for publication after addressing the following issues.

Point 1: The manuscript should be polished in language.

Response 1: English language is edited by a native English speaker. All corrections are yellow highlighted in the revised manuscript.

Point 2: Electronic structures of elements/ions could be removed.

Response 2: The authors believe that the electronic structures of elements/ions could be useful for easier understanding of differences in electrical conductivity, so prefer to keep them.

Point 3: Equations/formulas should be numbered.

Response 3: According to the Reviewer 2 suggestion, equations/formulas are numbered in the revised manuscript.

Point 4: Some figures should be enlarged.

Response 4: We have uploaded all figures in acceptable format (TIFF) and with appropriate resolution. We believe that technical editing will be done by the Journal management after accepting the manuscript.

Reviewer 3 Report

In this work, the authors reported the electrical and dielectric properties of CCTO doped with Ru in the TiO2 sites. The novelty of this work is clear because, to the best of my knowledge, there is no report on the dielectric properties of this material (even though the dielectric properties of CCTO/RuO2 have ever been reported). However, there are many points that the authors must carefully consider revising clearly before consideration for publication in Materials.

In the Abstract, the important numerical results must be included in the Abstract such as the dielectric constant at 1 RT and 1 kHz, loss tangent, etc.

In the Introduction, the motivation of this work was not clear. The literature review on the relevant previous works must be re-written. The replacement of Ti4+ with isovalent ions should be mentioned. The dielectric properties of isovalent dopant like Ru4+ must be mentioned and reviewed in the Introduction such as Sn4+ (https://doi.org/10.1016/j.jeurceramsoc.2021.04.017), Zr4+ (https://doi.org/10.1016/j.jallcom.2013.01.090;),Ge4+ (https://doi.org/10.1016/j.jallcom.2021.160322), etc.

As well known, the dielectric properties of CCTO ceramics are related to their microstructure. Thus, more details of microstructure analyses should be represented such as the SEM images of the sintered ceramics, the mean grain size, relative densities of the sintered ceramics.

Fig. 9, y-scale must be optimized to clearly represent the overall dielectric constant. The maximum scale may be 20k.

Table 5, the authors showed the dielectric constant, but not include the dielectric loss tangent.

The possible mechanism of a large value of conductivity / loss tangent must be explained. This result was likely related to the percolation effect like in the previous work for CCTO/RuO2 composites (http://dx.doi.org/10.1063/1.4893009)?

Author Response

We are very thankful to the reviewers for this thorough review. We have revised our manuscript according to the Reviewer's useful suggestions and comments.

In this work, the authors reported the electrical and dielectric properties of CCTO doped with Ru in the TiO2 sites. The novelty of this work is clear because, to the best of my knowledge, there is no report on the dielectric properties of this material (even though the dielectric properties of CCTO/RuO2 have ever been reported). However, there are many points that the authors must carefully consider revising clearly before consideration for publication in Materials.

Point 1: In the Abstract, the important numerical results must be included in the Abstract such as the dielectric constant at 1 RT and 1 kHz, loss tangent, etc.

Response 1: We agree with the Reviewer 3 that some important numerical results missing in the Abstract. However, allowed number of words in the Abstract is 200 which we already reached. To include new data, it is necessary to completely rewrite the Abstract and exclude some of presented ones. If all three Reviewers and the Editor consider it to be necessary, we will do it. 

Point 2: In the Introduction, the motivation of this work was not clear. The literature review on the relevant previous works must be re-written. The replacement of Ti4+ with isovalent ions should be mentioned. The dielectric properties of isovalent dopant like Ru4+ must be mentioned and reviewed in the Introduction such as Sn4+ (https://doi.org/10.1016/j.jeurceramsoc.2021.04.017), Zr4+ (https://doi.org/10.1016/j.jallcom.2013.01.090;), Ge4+ (https://doi.org/10.1016/j.jallcom.2021.160322), etc.

Response 2: In the revised manuscript the Introduction is rewritten and the replacement of Ti4+ with isovalent cations such as Sn4+, Zr4+, Ge4+ is emphasized while three new references are added in the Introduction and in the Reference list.

Added references:

  1. Li, M.; Cai, G.; Zhang, D.F.; Wang, W.Y.; Wang, W.J.; Chen, X.L. Enhanced dielectric responses in Mg-doped CaCu3Ti4O12, J. Appl. Phys. 2008, 104, 074107. doi: 10.1063/1.2989124
  2. Boonlakhorn, J.; Chanlek, N.; Srepusharawoot, P.; Thongbai, P. Improved dielectric properties of CaCu3-xSnxTi4O12 ceramics with high permittivity and reduced loss tangent, J. Mater. Sci. Mater. Electron. 2020, 31, 15599–15607. doi: 10.1007/s10854-020-04123-x
  3. Boonlakhorn, J.; Thongbai, P.; Putasaeng, B.; Kidkhunthod, P.; Maensiri, S.; Chindaprasirt, P. Microstructural evolution, non-Ohmic properties, and giant dielectric response in CaCu3Ti4-xGexO12 ceramics, J. Am. Ceram. Soc. 2017, 100, 3478–3487. doi: 10.1111/jace.14886
  4. Xu, D.; Zhu, Y.; Zhang, B.; Yue, X.; Jiao, L.; Song, J.; Zhong, S.; Ma, J.; Bao, L.; Zhang, L. Excellent dielectric performance and nonlinear electrical behaviors of Zr-doped CaCu3Ti4O12 thin films, J. Mater. Sci.: Mater. Electron. 2018, 29, 5116–5123. doi:10.1007/s10854-017-8475-0
  5. Xu, Z.; Qiang, H. Enhanced dielectric properties of Zn and Mn co-doped CaCu3Ti4O12 ceramics, J. Mater. Sci. Mater. Electron. 2017, 28, 376–380. doi: 10.1007/s10854-016-5533-y
  6. Xu, Z.; Qiang, H.; Chen, Y.; Chen, Z. Microstructure and enhanced dielectric properties of yttrium and zirconium co-doped CaCu3Ti4O12 ceramics, Mater. Chem. Phys. 2017, 191, 1–5. doi: 10.1016/j.matchemphys.2017.01.015
  7. Boonlakhorn, J.; Chanlek, N.; Manyam, J.; Srepusharawoot, P.; Thongbai, P. Simultaneous two-step enhanced permittivity and reduced loss tangent in Mg/Ge-Doped CaCu3Ti4O12 ceramics, J. Alloys Compd. 2021, 877, 160322. doi: 10.1016/j.jallcom.2021.160322

In the revised manuscript number of the references in the Reference list is 38 instead 31.

The aim of our work was to optimize the amount of ruthenium in the CaCu3Ti4-xRuxO12 crystal structure in order to obtain inexpensive and commercially acceptable materials with the desired functional properties – i.e. to synthesize compound isostructural with CaCu3Ti4O12 but with much larger conductivity which can be successfully used as interlayer between dielectric ceramic and metallic electrode with the aim to reduce interlayer stress.

To be more precise, one sentence is added in the revised manuscript “Quite contrary, our interest was to synthesize CaCu3Ti4-xRuxO12 as compound isostructural with CaCu3Ti4O12 but with much larger conductivity which can be successfully used as interlayer between dielectric ceramic and metallic electrode with the aim to reduce interlayer stress.”

Point 3: As well known, the dielectric properties of CCTO ceramics are related to their microstructure. Thus, more details of microstructure analyses should be represented such as the SEM images of the sintered ceramics, the mean grain size, relative densities of the sintered ceramics.

Response 3: We agree that microstructure influences dielectric properties of material. The results related to the effect of sintering to CCTO ceramics microstructure and dielectric properties we published earlier in our paper: Marković, S., Lukić, M., Jovalekić, Č., Škapin, S.D., Suvorov, D., Uskoković, D. Sintering Effect on Microstructure and Electrical Properties of CaCu3Ti4O12 ceramics. Ceram. Trans. 2013, 240, 337–348, doi: 10.1002/9781118744109.ch37.

In our future research we are planning to examine an influence of different sintering conditions (different heating rate, final temperature, dwell time and sintering atmosphere – air and argon) on dielectric properties of CaCu3Ti4O12, CaCu3Ti3RuO12, and CaCu3Ti4O12/CaCu3Ti3RuO12 functionally graded materials. That is why in this manuscript microstructure of sintered samples was not included.

As we stated in the Introduction the primary goal of this study is to show that obtaining of commercially acceptable materials with the desired functional properties is possible by substation of one Ru ion in CCTO:

For the fabrication of capacitors with the good capacitive performance, it is important to combine dielectric ceramic and electrodes with similar crystal structures and unit cell parameters. The reduction of the stress on the ceramic-electrode interfaces can be achieved by using commercially available materials as an interlayer having a close lattice parameter match with both dielectric and electrode. It was noticed that the incorporation of Ru4+ ions in the CaCu3Ti4O12 crystal structure dramatically increases the conductivity of these materials. The CaCu3Ru4O12 materials are isostructural with CaCu3Ti4O12 materials with cubic space group showing the Pauli-paramagnetic and metallic character. With the CaCu3Ru4O12 material as an interface between CaCu3Ti4O12 ceramic and metallic electrode, it is possible to reduce the stress on the dielectric-electrode interfaces.

Since ruthenium in an expensive chemical element, the aim of our work was to optimize the amount of ruthenium in the CaCu3Ti4-xRuxO12 crystal structure in order to obtain inexpensive and commercially acceptable materials with the desired functional properties – i.e. to synthesize compound isostructural with CaCu3Ti4O12 but with much larger conductivity.

Point 4: Fig. 9, y-scale must be optimized to clearly represent the overall dielectric constant. The maximum scale may be 20k.

Revised 4: According to shared suggestion of the Reviewers 1 and 3 Figure 9 is revised; the y-axis is rescaled to 2x104.

Point 5: Table 5, the authors showed the dielectric constant, but not include the dielectric loss tangent.

Response 5: Table 5 shown in the revised manuscript contains both the dielectric constant and the dielectric loss tangent values.

Point 6: The possible mechanism of a large value of conductivity / loss tangent must be explained. This result was likely related to the percolation effect like in the previous work for CCTO/RuO2 composites (http://dx.doi.org/10.1063/1.4893009)?

Response 6: In the paper “Enhancement of high dielectric permittivity in CaCu3Ti4O12/RuO2 composites in the vicinity of the percolation threshold”, by Rupam Mukherjee, Gavin Lawes, and Boris Nadgorny, published in APPLIED PHYSICS LETTERS 105, 072901 (2014), the authors examined composite nanoparticle system consisting of metallic RuO2 grains embedded into CaCu3Ti4O12 matrix while we examined powders which are a single-phase perovskite oxide without a secondary phase.

The aim of Mukherjee, Lawes and Nadgorny, was to increase the dielectric constant of CCTO by introduction of RuO2, and production of RuO2/CCTO composites. Quite contrary, our aim was to enhance conductivity.

As we stated in the manuscript, different conductivity between the CCTO and CCT3RO samples can be explained by the Kondo mechanism:

The difference in electrical conductivity can be explained by different electron configurations of Ti4+ and Ru4+ ions. Titanium is a 3d element with an electronic configuration:

Ti: 1s2 2s2 2p6 3s2 3p6 4s2 3d2

However, the electronic configuration of Ti4+ cation in the CCTO crystal structure turns into:

Ti4+: 1s2 2s2 2p6 3s2 3p6

with closed external 3p6 shells which explain the relatively low conductivity of these materials.

The isomorphic replacement of the Ti4+ ions by Ru4+ leads to different electronic configurations. Explicitly, Ru4+ has the electronic configuration:

Ru4+: 1s2 2s2 2p6 3s2 3p6 4s2 3d10 4p6 5s0 4d4,

with uncompleted d subshells. These four electrons in the octahedral crystalline environment are situated in tg crystalline orbitals containing two unpaired d electrons, i.e., Ru4+ represents a magnetic ion in the CCRO crystal. The CCRO conductivity can be explained by the Kondo mechanism. Namely, CCRO materials are d-electron heavy-fermion system, which behaves similarly to an f-electron heavy-fermion system. In CCRO structures the Cu 3d electrons are localized, while 4d electrons of magnetic Ru4+ ions contribute to the conductivity.

The authors remark:

The authors noticed omission in the Arrhenius equation, page 9; the omission is corrected in the revised manuscript. Actually,  lnσ= σ0exp⟨-Ea/kBT⟩  is corrected to be σ=σ0exp⟨-Ea/kBT⟩  

Round 2

Reviewer 1 Report

The revised manuscript can be now accepted.

Reviewer 2 Report

The manuscript has been well revised. It is recommended for publication as it is.

Reviewer 3 Report

All comments have been revised. Accepted.